# *Fusarium* Species Associated with Diseases of Citrus: A Comprehensive Review

**DOI:** 10.3390/jof11040263

**Published:** 2025-03-28

**Authors:** Mihlali Badiwe, Régis Oliveira Fialho, Charles Stevens, Paul-Henri Lombard, Jan van Niekerk

**Affiliations:** 1Department of Plant Pathology, Stellenbosch University, Private Bag X1, Stellenbosch 7602, South Africa; regisf@sun.ac.za (R.O.F.); phlombard@sun.ac.za (P.-H.L.); janvn@cri.co.za (J.v.N.); 2Citrus Research International, P.O. Box 28, Nelspruit 1200, South Africa; charless@sun.ac.za

**Keywords:** species dry root rot, identification, molecular mechanism of resistance, pathogenic mechanism, resistance mechanisms of plants, mycoviruses, biocontrol agents

## Abstract

The citrus industry contributes to the cultivation of one of the most important fruit crops globally. However, citrus trees are susceptible to numerous *Bisifusarium*, *Fusarium*, and *Neocosmospora*-linked diseases, with dry root rot posing a serious threat to citrus orchards worldwide. These infections are exacerbated by biotic and abiotic stresses, leading to increased disease incidence. Healthy trees unexpectedly wilt and fall, exhibiting symptoms such as chlorosis, dieback, necrotic roots, root rot, wood discolouration, and eventual decline. Research indicates that the disease is caused by a complex of species from the Nectriaceae family, with *Neocosmospora solani* being the most prominent. To improve treatment and management strategies, further studies are needed to definitively identify these phytopathogens and understand the conditions and factors associated with *Bisifusarium*, *Fusarium*, and *Neocosmospora*-related diseases in citrus. This review focuses on the epidemiology and symptomatology of *Fusarium* and *Neocosmospora* species, recent advances in molecular techniques for accurate phytopathogen identification, and the molecular mechanisms of pathogenicity and resistance underlying *Fusarium* and *Neocosmospora*–citrus interactions. Additionally, the review highlights novel alternative methods, including biological control agents, for disease control to promote environmentally friendly and sustainable agricultural practices.

## 1. Introduction

The *Citrus* genus comprises flowering trees and shrubs in the Rutaceae family. *Citrus* spp. produce fruits that are among the most widespread and extensively produced fruit types in the world [1]. These include important crops such as oranges, lemons, grapefruits, mandarins, pomelos, and limes [2]. The crop is grown commercially in more than 140 countries around the world [3]. The contribution of the citrus industry to the world economy is significant, as it provides employment opportunities to millions of people in the production, handling, transportation, storage, and marketing sectors. The importance of citrus fruit is attributed to its diversified use, which is widely consumed either as a highly nutritious fresh fruit, as juice, or as canned, dried, or frozen fruit. The skin, pulp, and seeds constitute an exceptional repository of phytochemicals and biologically active compounds with health-promoting properties [4,5,6]. Furthermore, the fruits are also utilized for flavouring foods and beverages and as a raw material in various other industries, including biofuels, cosmetics, and medicine [7,8,9]. However, citrus trees are susceptible to a wide variety of phytopathogens, including bacteria, fungi, and viruses, giving rise to diseases at nearly all stages of citrus production [10].

*Fusarium* is among the most prominent genera in the kingdom fungi, incorporating a broad spectrum of morphologically and phylogenetically diverse fungi [11]. These fungi are mostly soilborne saprophytes organisms, which colonize both living and dead plant tissue as endophytes or epiphytes [12]. Due to the complex diversity of this genus, over time, it has undergone taxonomic revisions encompassing *Fusarium*-like genera such as *Bisifusarium* L. Lombard, Crous & W. Gams, *Fusarium dimerum* species complex (SC), *Neocosmospora* E.F. Sm. (*Fusarium solani* SC), and *Rectifusarium* L. Lombard, Crous & W. Gams (*Fusarium ventricosum* SC) [13,14].

*Fusarium* is historically a broad genus that encompassed a large variety of filamentous fungi recognized for their agricultural, ecological, and medical significance [15,16,17]. However, significant improvements in molecular phylogenetics have aided the understanding that many species within *Fusarium sensu lato* are not as closely related as initially thought. Thus, the genus has been divided into multiple genera, including *Fusarium*, *Neocosmospora*, *Bisifusarium*, and *Rectifusarium* [16,18]. This was an advancement because molecular studies using DNA sequencing (e.g., *ITS*, *TEF1*, *RPB1*) revealed phylogenetic distinctions by demonstrating significant genetic differences among species previously grouped under *Fusarium* [18]. Furthermore, *Fusarium sensu lato* was found to be polyphyletic; thus, dividing it into monophyletic groups, each with a single evolutionary origin, aligned taxonomy with evolutionary history [16,17]. Thus, separating them assists in clarifying their ecological and pathogenic roles, as they occupy diverse ecological niches, such as plant pathogens (*Fusarium oxysporum*), saprobes (*Neocosmospora*), or opportunistic pathogens of humans [15].

*Fusarium* and *Neocosmospora* spp. have significant implications for the agricultural sector and food safety, causing devastating diseases in a wide range of crops, including cereals, fruits, and vegetables. Moreover, these genera contain species that are capable of mycotoxin production (Table 1), which are among the most pivotal factors contributing to food loss, food safety challenges, and plant death [19,20]. *Fusarium* spp. produce a wide range of mycotoxins that contaminate food, thereby reducing its quality and posing a threat to the health of citrus trees, as well as to both human and animal consumers. Furthermore, mycotoxin contamination contributes to poor product quality and loss of export revenue, resulting in significant economic losses and increased malnutrition [21,22].

These genera are implicated in diverse and impactful diseases on many crops, including citrus. In the case of citrus, the resulting diseases encompass decline, dieback, dry root rot (DRR), citrus fruit rot, twig blight and rot, vascular wilt, and root and stem rot [23,24,25,26,27]. Despite the frequency of disease incidence, our understanding of citrus fungal diseases caused by *Fusarium* and *Neocosmospora* species remains limited. Further research and exploration are crucial to enhancing our comprehension of these pathogens and developing effective management strategies. Furthermore, an extensive record of reported investigations linking *Fusarium* and *Neocosmospora* spp. to devastating citrus disease exists, whereby countries like Brazil, the Canary Islands, China, Egypt, Greece, India, Iraq, Italy, Morocco, Oman, Pakistan, South Africa, and the USA, have all confirmed the presence of disease-causing *Neocosmospora solani* in their citrus orchards [11,28,29,30,31,32,33,34,35,36,37]. Furthermore, *N. solani* was isolated from the bark and roots of sweet lime (*C. limon* Limetta) and key lime (*C. aurantifolia*), displaying symptoms of wilting [38]. Table 2 summarizes recent incidences of *Neocosmospora* and *Fusarium*-related diseases in citrus around the world.

Comprehending and investigating the effects of *Fusarium* and *Neocosmospora* diseases on citrus orchards is crucial to ensuring the sustainable management of their associated diseases and, consequently, the success of the citrus industry. This knowledge not only enhances the resilience of citrus orchards against existing threats but also establishes a basis for taking proactive measures in response to emerging challenges. This review, therefore, seeks to offer insights into recent advancements in the identification of *Fusarium* and *Neocosmospora* spp. and their pathogenic mechanisms, the molecular mechanisms of resistance in citrus, and the potential utilization of mycoviruses, bacteria, and fungi as biocontrol agents against *Fusarium* diseases of citrus.

**Table 1 jof-11-00263-t001:** Concise overview of relevant *Fusarium* species, their mycotoxins, and how they impact citrus.

*Fusarium* Species	Mycotoxin(s)	Impact on Citrus	Other Affected Crops	Reference
*Neocosmospora solani*	Fusaric acid	Contributes to dry root rot and decline and impairs plant root health, reducing nutrient uptake and chlorosis, and causing overall decline in tree health.	Various fruit and vegetable crops	[39]
*Fusarium* *oxysporum*	Beauvericin, Enniatins	Contributes to wilt and root diseases and compromises vascular health, leading to blockage, leaf yellowing, and tree death.	Bananas, cotton, tomatoes, melons	[40]
*Fusarium* *proliferatum*	Fumonisins, Moniliformin	Rare in citrus but can cause fruit decay. Contributes to plant stress and reduced quality.	Asparagus, garlic, maize, onions rice	[41]
*Fusarium* *verticillioides*	Fumonisins	No direct reports of citrus infection. Contributes to seedling diseases, root rot, and reduced growth.	Maize (primary host), wheat, rice	[42]
*Fusarium* *graminearum*	Zearalenone, Trichothecenes	Not typically associated with citrus; contributes to ear rot, grain contamination, and reduced crop yield in cereals.	Barley, maize, wheat	[43]

**Table 2 jof-11-00263-t002:** Recent incidences of *Fusarium*-related diseases in *citrus* species worldwide.

Citrus Host	*Fusarium* Species	Disease	Symptoms	Country	References
*Citrus reticulata*	*F. oxysporum* *F. equiseti*	*Fusarium* rot	Stem end discolouration and fruit rot	Pakistan	[39,43]
*Poncirus trifoliata*	*F. oxysporum* *F. solani*	*Fusarium* root rot	Leaf yellowing, wilting, and decline	China	[44]
*Citrus* spp.	*N. citricola* *N. ferruginea* *N. solani*	Dry root rot	yellowing, wilting leaves, and dieback cracked trunks above the crowns	South Africa	[45]
*Citrus* spp.	*F. solani* *F. oxysporum* *F. equiseti* *F. brachygibbosum*	Dry root rot	Root rot Necrotic roots Purple wood discoloration Plant yellowing	Morocco	[25]
*Citrus sinensis*	*Fusarium* spp. *F. polyphialidicum*	Vascular wilt	Chlorosis, defoliation, and wilting of branches	Mexico	[46]
*Citrus unshiu* *Citrus aurantium*	*N. solani*	Dry root rot	Light purple, vascular discolouration, and dry decay of fibrous roots	Turkey	[47]
*Poncirus trifoliata*	*N. solani* *F. oxysporum*	Dry root rot	Chlorosis, canopy reduction, wilting, root necrosis, defoliation, and plant death symptoms	Chile	[48]

## 2. *Fusarium* Species Associated with Citrus

### 2.1. Taxonomy and Diversity of Fusarium in Citrus Trees

*Fusarium* species are acknowledged as a major group of pathogens causing notable symptoms in citrus trees, including DRR, vascular wilt, root and stem rot, twig rot, dieback, and twig blight [44,45,48,49]. Moreover, citrus fruit decay has been associated with various *Fusarium* and *Neocosmospora* species, such as *N. solani*, *F. oxysporum*, and *F. concentricum* [39,48,50]. Additional pathogens, such as *F. proliferatum* and *F. sambucinum,* have also been linked to the disease [30]. The specific roles of *F. equiseti* and *F. semitectum* in disease development remain uncertain despite their presence in citrus roots, as they are generally regarded as saprophytic colonizers [51]. Several *Fusarium* species, such as *F. oxysporum*, *F. sambucinum*, and *N. solani*, have been documented in various regions like Florida and Greece [30]. In Tunisia, *F. oxysporum* f. sp. *citri* has been recognized as the causal agent of wilt in citrus trees [52]. Recently, the taxonomy of *Fusarium* has undergone refinement, resulting in its segregation into *Fusarium*-like genera such as *Neocosmospora* (previously included in the *Fusarium solani* species complex), *Bisifusarium* (previously classified as the *F. dimerum* SC), and *Rectifusarium* (previously assigned to the *F. ventricosum* SC) [14]. Recent studies have identified novel species linked to the disease, including *F. ensiforme*, *F. siculi*, *N. croci*, *N. macrospora*, and *F. brachygibbosum* [11,53]. Furthermore, strains initially classified as *F. ensiforme* and subsequently reclassified as *N. brevis* have been linked to DRR in Italy, along with other *Neocosmospora* species in Europe [11].

### 2.2. Modern Molecular Techniques for Accurate Identification of Pathogens

The emergence of molecular techniques has introduced a variety of strategies for identifying species associated with citrus DRR, offering distinct advantages over conventional detection methods [27]. These modern molecular methods include conventional polymerase chain reaction (PCR), real-time PCR (RT-PCR), and DNA sequencing, recognized for their enhanced sensitivity, reliability, specificity, and reproducibility [54,55]. PCR-based investigations facilitate the in vitro amplification of specific segments of target nucleic acids using primers designed for specific species. Numerous studies have employed species-specific primers for detecting *Fusarium* spp. linked to DRR, utilizing sequences from ITS-rDNA subunit *TEF-1α* and the Calmodulin gene for this purpose [56,57,58]. Kurt [47] utilized sequencing of *ITS* and *TEF-1α* nucleotides, pathogenicity assays, and MALDI-TOF MS to characterize the pathogenicity of *N. solani* associated with dry root rot of citrus in the eastern Mediterranean region of Turkey.

The genus *Fusarium* exhibits significant diversity in morphology, phylogeny, and host specificity, which presents challenges when distinguishing between *Fusarium*-like species. Therefore, to assess *Fusarium* diversity, various universal fungal primers designed for amplifying multi-locus genes have been developed for phylogenetic analysis [27]. The internal transcribed spacer region of ribosomal DNA (ITS-rDNA), a gene present in multiple copies within fungal genomes, offers extensive variability between genera, making it a prominent target for molecular fungal identification [59]. However, solely relying on the internal transcribed spacer (ITS) region for fungal identification offers significant constraints, particularly in distinguishing closely related species. The ITS region’s variability is inconsistent across different fungal groups, which can yield inaccurate identifications for certain taxa [60,61]. To overcome these challenges, multi-locus sequencing approaches have been developed, integrating additional genetic markers such as translation elongation factor 1-alpha (*TEF1*), RNA polymerase II subunits (*RPB1* and *RPB2*), and beta-tubulin. These multi-locus approaches provide pronounced resolution by attaining genetic variation that ITS alone may not discover, thus making it possible to attain more accurate species delimitation and robust phylogenetic analyses [62,63]. Consequently, multi-locus approaches have become standard in fungal taxonomy, facilitating deeper insights into fungal diversity and evolutionary relationships [64,65].

By employing molecular techniques, over 300 species and 22 species complexes were identified and characterized, constituting a diverse array of phytopathogenic species [12]. The MycoBank database currently documents more than 1400 names associated with *Fusarium* [11]. With the advent of modern molecular tools, systematic classification has been established for the majority of phytopathogenic *Fusarium* spp. sequencing for various DNA loci aids in discriminating between taxa. In the realm of *Fusarium* and *Fusarium*-like species, the utilization of *EF-1α* and *RPB2* and other regions, such as those encoding nuclear and mitochondrial rRNA, *calmodulin* (*CAM*), *β-tubulin* (*TUB*), and *histone H3* loci, have proven valuable for identifying all pathogenic species [11,62]. Moreover, *RPB2* loci, especially in the case of FSSC, exhibit high resolving power, enabling accurate delineation of pathogenically relevant clades and ensuring precise identification of all pathogenic species [48,52].

### 2.3. Fusarium and Neocosmospora Species and Their Impact

Investigations suggest that DRR disease in citrus trees arises from a complex involving various *Fusarium* and *Fusarium*-like species. It has been outlined that *N. solani*, classified separately from the *Fusarium* genus, is significantly associated with DRR [52]. In a comprehensive study conducted in Italy, the authors identified various *Fusarium*-like species linked to citrus plantations [52]. They isolated distinct species, including *Fusarium* and *Neocosmospora* spp., from different parts of symptomatic citrus trees such as roots, trunk, branches, and twigs. These species encompass *F. sarcochroum*, *F. oxysporum*, *F. ensiform*, *N. solani*, and several new species. Three of these species belong to the genus *Fusarium*, specifically *F. citricola* and *F. salinense* in the *F. citricola* complex, and *F. siculi* in the *F. fujikuroi* complex, while the remaining two belong to the genus *Neocosmospora* (*N. croci* and *N. macrospora*). *Fusarium salinense* has been implicated in galls of *C. sinensis* in Sicily and the Aeolian Islands, and *F. citricola* has been identified in galls affecting several citrus species [52]. *Fusarium siculi* has been isolated from symptomatic crowns of *C. sinensis*, although its definitive pathogenicity has not yet been conclusively established. Furthermore, *F. sarcochroum*, documented in Greece in the 1970s [63], was later isolated from wilted lemon and mandarin branches in Italy in 2018 [52].

Identification of *Fusarium* spp. linked to plant diseases relies on both morphological characteristics and sequencing of the internal transcribed spacer (ITS) region. A study conducted in South Africa on citrus trees affected by DRR identified several species of *Neocosmospora* associated with the disease [45]. These species, including *N. solani*, *N. citricola*, *N. ferruginea*, *N. brevis*, *N. crassa*, *N. hypothenemi*, *N. noneumartii*, *N. addoensis*, *N. gamtoosensis*, *N. lerouxii*, and *N. falciformis*, were collected from the tree canopy, trunks, roots, and soil in citrus groves where the disease had been recorded. The first study identifying *F. equiseti* as the causative agent responsible for *Fusarium* rot in mandarin (*C. reticulata* ‘Kinnow’) was conducted by performing DNA extractions, PCR analysis, and sequencing of isolates obtained from fruit displaying symptoms of the disease [46]. In the first report on molecular identification of *Fusarium* spp. causing fruit rot in mandarin in Bangladesh, a 99.42% similarity of the sequence obtained from isolated fungi with the reference *F. concentricum* was found [50]. Similarly, *F. tricinctum* was first reported as causing *Fusarium* fruit rot in Navel orange (*C. sinensis*) in China in a study conducted in 2023 [63].

## 3. Etiology and Symptomatology of *Fusarium* spp. Associated with Citrus

### 3.1. Etiology of the Associated Diseases

The infection process of *Fusarium* and *Neocosmospora* spp. in citrus can be broken down into several key stages, beginning with the persistence of spores in the soil (Figure 1). *Fusarium* spp. produce conidia, which are asexual spores, in specialized structures known as sporodochia, typically formed on infected roots or plant debris. These conidia, along with other spores such as chlamydospores, persist in the soil due to their thick-walled, durable nature, which makes them well adapted for survival in nonoptimal conditions, such as fluctuating moisture levels and temperature extremes. Once the environment becomes favourable, such as in the presence of moisture and suitable temperatures, the spores begin to germinate. During germination, the spores develop hyphal growth that forms an extensive mycelial network. This network spreads through the soil, facilitating the fungus’s ability to infect nearby plants or newly planted citrus trees. As the mycelium grows, specialized fungal structures called conidiophores are formed. These conidiophores produce new conidia, which are then released into the soil, where they are capable of infecting new roots. The spread of conidia through the soil continues the cycle of infection. When the hyphal network encounters citrus roots, the fungus adheres to the root structures and begins the infection process. The hyphae penetrate the root epidermis through natural openings, such as lenticels or root tips, or through injuries, using mechanical pressure and enzymatic activity to facilitate penetration. Once inside the roots, the fungus colonizes the plant by growing through the vascular tissue. This disruption of the plant’s vascular system impairs its ability to transport water and nutrients, leading to root rot, wilting, and other symptoms of decline. The spread of the fungus within the root system further exacerbates the damage, making it harder for the plant to recover, thus increasing the severity of the infection [17,64,65].

Several studies have highlighted the association between the presence of the citrus nematode *Tylenchulus semipenetrans* and the exacerbation of DRR in citrus rootstocks. This correlation is associated with a noticeable impact on the plant’s growth response, leading to a reduction in plant height [31,32]. The authors of these studies emphasized that the synergistic influence of *N. solani* and *T. semipenetrans* may contribute to the increased incidence of *Fusarium* root rot on citrus rootstocks that exhibit tolerance to *Fusarium*. Other pathogens, including nematodes such as *T. semipenetrans* [31] and specific fungi such as *Neonectria macrodidyma* Halleen, Schroers & Crous (*Cylindrocarpon macrodidymum*), isolated from symptomatic citrus trees in California, are believed to play a role in DRR disease progression [66]. Furthermore, it has been reported that inoculation of citrus trees with *N. solani* and *P. nicotianae* or *P. citrophthora* Leonian resulted in greater root rot compared to inoculation with *P. nicotianae* or *P. citrophthora* alone [67].

### 3.2. Symptomatology

*Fusarium* and *Neocosmospora* spp. are known to cause various diseases in citrus trees and fruits, each exhibiting similar yet distinct symptoms and severities. These species present substantial risks by causing diseases such as fruit rot, root rot, seedling damping-off, and plant wilt in numerous economically significant crops globally, encompassing citrus, vegetables, ornamentals, and field crops [68,69,70]. The infections are intricately linked to the interactions between the host and pathogens, as well as the presence of biotic and/or abiotic stresses, ultimately influencing disease incidence and expression [27]. They depend on plants for nutrients and have evolved mechanisms allowing them to colonize the plant tissues [71,72]. *Fusarium* wilt in citrus typically manifests as wilting and yellowing of leaves, eventually leading to their death, accompanied by a dark discolouration observable in the vascular tissue. Severe wilting attributed to *F. oxysporum* has been documented in *C. sinensis* ‘Washington Navel’ and *C. clementina* ‘Cassar’, and ‘Hernandina’ from commercial Tunisian orchards in the CapBon area, with noticeable discolouration of vascular tissue [52]. Additionally, wilting, chlorosis, and epinasty of young leaves, along with necrosis of rootlets, large roots, and crowns, have been observed in sour orange seedlings and orange trees (*C. sinensis*) var. ‘Valencia’ [52,73,74].

The progression of citrus DRR initiates with fungal penetration into the roots, leading to root rot, which then progresses into the xylem vessels, causing tree weakening. Typical symptoms include yellowing along the central vein of leaves, chlorosis, necrotic roots, root rot, purple wood discolouration, and dieback, ultimately culminating in tree subsidence, wilting, and eventual death (Figure 2) [11,24,27]. Investigations suggest that this disease is primarily caused by a complex of *Fusarium* spp., and *N. solani* being the most prevalent [27]. Infected trees often suffer crown loss due to desiccation, leaf fall, and fruit drying on branches [75]. Additionally, fibrous root rot is evident, correlated with reductions in crown size, defoliation, root cortex scaling, and, ultimately, tree death [76]. In some instances, hemiplegia along the longitudinal axis may occur in affected trees [30]. Citrus DRR is complex due to various biotic factors such as *Phytophthora* spp., citrus tristeza virus (CTV), *T. semipenetrans*, rodents, and insects, as well as abiotic stresses including drought, high temperatures, changes in soil pH, over-fertilization, root asphyxiation, and poor root aeration [77,78,79,80]. These factors collectively create opportune conditions for *Fusarium* and *Neocosmospora* spp., making it challenging to predict the development of citrus DRR. Further investigations are imperative to accurately identify the causative pathogens and understand the critical conditions and factors associated with citrus DRR occurrence.

## 4. *Fusarium* spp.—Host Plant Interaction and Molecular Mechanisms

### 4.1. Host–Pathogen Interactions

The interactions between *Fusarium* spp. and host plants are intricate and multifaceted, often implying morphological and physiological alterations in both entities [71]. The *Fusarium* spp. pathogenic arsenal responsible for host plant invasion and disease includes host-specific toxins, effector proteins, and cell-wall-degrading enzymes. *Fusarium* spp. display high host specificity and effectively colonize and damage plant tissues through the production of host-specific toxins, enabling the pathogen to evade plant defence mechanisms (Figure 3) [81]. Upon encountering a host, *Fusarium* spp. trigger a complex network of interconnected signalling pathways. Many phytopathogenic *Fusarium* spp. produce toxins (Table 1) that induce plant disease by increasing host plant cell membrane permeability and electrolyte leakage, resulting in membrane damage, disrupted physiological processes, and eventual cell death [82,83]. These toxins, characterized by low molecular weight, manifest distinct symptoms in host plants such as chlorosis, growth inhibition, leaf spotting, necrosis, and wilting [84]. Additionally, they compromise the integrity of chloroplast inner membranes, leading to progressive basal lamella breakdown [81,85]. Moreover, these toxins can target host plant mitochondria, disrupting mitochondrial membrane structure and causing cristae swelling, vacuolization, and damage, ultimately impairing mitochondrial function [83]. *Neocosmospora solani* is recognized for its capacity to synthesize and release certain mycotoxins, contributing to its pathogenicity by inducing plasmolysis in vascular parenchyma cells at the root level and subsequent translocation to the leaves [86,87]. Throughout the infection process, the production of mycotoxins like naphthazarin has been documented, causing symptoms such as vein discolouration, leaf wilting, and the obstruction of vascular transport vessels [88,89].

### 4.2. Effector Proteins

Additionally, the process of infection is facilitated by secondary metabolites, including effector proteins, produced by *Fusarium* spp., which prompt the host plant to initiate a defence response known as effector-triggered immunity (ETI) or pathogen-associated molecular pattern (PAMP)-triggered immunity [90]. These effector proteins, such as Fg62 and Avr2, play a significant role in plant cells, thus influencing the host–phytopathogen interaction [91]. Successful infection relies on numerous convoluted biochemical pathways, including mitogen-activated protein (MAP) kinase signalling pathways [92], Ras proteins [93], G-protein signalling components [69,94,95], the velvet complex components [96], and cAMP pathways [93] (Figure 3). Several studies have investigated plant-phytopathogenic fungal effectors, particularly those with available whole-genome data, with *F. oxysporum* f. sp. *lycopersici* being a popular source of cloned phytopathogenic effectors [97,98]. Consequently, the virulence effects and transport molecular mechanisms of effector proteins remain largely unknown, given the infancy of research in this field. The specific effector proteins responsible for modifying plant metabolism to ensure nutrient availability for the infecting phytopathogen are yet to be identified, as are the plant signal transduction pathways that control effector protein gene expression [83]. However, comparative analysis may identify common motifs present in effectors of phytopathogenic fungi. Acquiring this knowledge and understanding would be crucial for elucidating the mechanisms involved in phytopathogen–plant host interactions, revealing the dynamic pathogenic mechanisms of *Fusarium* spp. and the host plant disease-resistance mechanisms [83].

### 4.3. Cell-Wall-Degrading Enzymes

Phytopathogenic fungi, including *Fusarium* spp., secrete cell-wall-degrading enzymes such as cutinases, cellulases, lipases, pectinases, and xylanases, which play a crucial role in successful penetration of the plant cell wall [99,100,101,102]. These enzymes facilitate the biochemical mechanisms involved in breaking down the cuticle and cell wall of the host plant during *Fusarium* spp. invasion, colonization, nutrient extraction, and proliferation [103,104]. The study of phytopathogenic cell-wall-degrading enzymes is currently a prominent research area, employing molecular biology and proteomics approaches. It has been documented that during infection, *F. graminearum* secretes cellulase, pectinase, and xylanase, which degrade host plant cell wall constituents and promote penetration and proliferation of the phytopathogen within the host tissue [105]. *Fusarium* spp. also produce β-galactosidases, which enhance the breakdown of lactose, producing galactose and glucose, thereby aiding in fruit softening [106]. High expression of β-galactosidase in the initial stages of fruit softening facilitates the degradation of cell wall galactosyl bonds, significantly reducing the integrity of the host plant’s cell wall [107]. Enzymes such as amylase, hemicellulase, phospholipase, and protease, produced during the infection process, have been reported to facilitate the degradation of important cell wall components such as starch, hemicellulose, lipids, and proteins, respectively [105]. The cellulase and pectinase enzymatic activity of *N. solani* has been reported to facilitate effective penetration across the entire root surface [108]. Moreover, *F. oxysporum* not only synthesizes cutinase and pectinase but also secretes enzymes like polygalacturonase, which degrade the cell wall to facilitate penetration [109].

It is important to note that cell-wall-degrading enzymes from phytopathogenic fungi, including *Fusarium* spp., are not the sole causative agents of infection and disease in host plants; coordinated secretion of phytohormones such as salicylic acid and jasmonic acid also plays a role in transducing the activation of the host plant’s defence systems against pathogen attack [110]. The interaction between *Fusarium* spp. and the host plant is biochemically complex, involving prompt expression of phytopathogen genes as well as activation of the plant’s enzymatic defence systems. This results in the formation of RNA interference, transcription of proteases and their inhibitors, biosynthesis of antifungal compounds, and production of reactive oxygen species (ROS) [111,112]. Collectively, these molecules drive a well-executed response to phytopathogen infections [83]. Therefore, effective management of *Fusarium* spp. diseases necessitates an in-depth understanding of the molecular mechanisms involved in the pathogenesis process. Resistant plants attempt to hinder the fungi’s expansion by synthesizing antifungal structures and compounds such as tyloses, gels, and gums to impede fungal growth and dissemination [71]. Citrus plants experience wilting because of xylem obstructions occurring either at the root or trunk levels. This results in the formation of a compact plug composed of interwoven filaments coated with an inert material at the intersection of xylem vessels [27]. DRR-affected citrus trees display an aggregation of unsaturated lipids, phospholipids, and choline at the plasmodesmata level. The onset of this accumulation is triggered by the progression of lipids through vessel pits from vessels into neighbouring parenchyma cells [28,113].

### 4.4. Induced Systemic Resistance (ISR) in Citrus

Induced Systemic Resistance (ISR) and Systemic Acquired Resistance (SAR) are essential for citrus plant defence against a wide range of pathogens. ISR is activated by beneficial microbial interactions or infection, enhancing the plant’s barriers without directly targeting the pathogens [114,115]. ISR triggers a salicylic-dependent signalling cascade, leading to systemic, broad-spectrum resistance. SAR, initiated by non-pathogenic stimuli such as mechanical wounding or herbivory, involves signalling molecules like salicylic acid (SA), jasmonic acid (JA), and ethylene, offering systemic protection against diverse pathogens [116,117]. These mechanisms can operate concurrently, providing cumulative pathogen protection [118]. Citrus plants can develop SAR through various treatments, including chemical elicitors, heat shock, or wounding, effectively managing diseases like citrus canker and Huanglongbing (HLB) [119,120]. SAR inducers, such as imidacloprid, thiamethoxam, and acibenzolar-S-methyl (ASM), have been shown to mitigate HLB symptoms in citrus trees [95,121]. Furthermore, the role of rootstock and rhizobacteria in inducing SAR against pests and pathogens highlights the importance of microbial interactions in plant defence [118,122,123].

Beneficial microbes, including plant growth-promoting bacteria and fungi, activate ISR, protecting against a broad spectrum of pathogens. Microbial strains such as *Bacillus amyloliquefaciens* and *T. harzianum* have been identified as key in activating defence pathways [124,125]. The application of beneficial fungi enhances defence against *Fusarium* root rot, with methods like root dipping and soil drenching proving effective [126,127,128,129]. Furthermore, the resistance of citrus rootstocks to *Fusarium* spp. and the application of fungal elicitors suggest methods to enhance citrus immunity [27,130]. A strain of *Xanthomonas citri* ssp. *citri* inducing a hypersensitive response in citrus indicates an alternative immunity mechanism against *Fusarium* spp. [131,132]. This examination underscores the multifaceted approach to defending citrus plants against *Fusarium* spp., combining microbial interactions and genetic resilience. By understanding and harnessing these natural defence mechanisms, new strategies for citrus disease management can be developed, reducing reliance on chemical control and enhancing sustainability in citrus agriculture.

### 4.5. Phytochemical Defence Mechanisms

Citrus plants harness phytochemicals as a multifaceted defence mechanism against diseases and herbivores, employing both direct and indirect strategies. These phytochemicals are crucial to the plant’s defence mechanism, which depends on genetic and environmental factors [133]. Notably, compounds like limonin, flavonoids, geranoxycumarine, and saponins have been pinpointed for their efficacy against *Fusarium* spp., which poses a significant threat to citrus crops [134]. These have been identified in citrus rind and extracts, and they possess antifungal potential, which underscores their value as natural fungicides, marking a pivotal step in the use of phytochemicals for plant protection [134,135,136]. More research into the phytochemistry of citrus fruits shows that they contain antifungal substances such as naringin, hesperidin, and especially limonin, which are very important for stopping the growth of bacteria and fungi [5,137,138,139]. While the precise mechanisms of their action remain under study, these discoveries hint at promising applications as natural fungicidal agents, offering a sustainable option for the citrus industry [136,140].

Enhancing the intricate defence network of citrus plants, specific phytochemicals such as citral activate defence pathways, including the jasmonic acid pathway, enhancing pathogen resistance. This involves the induction of plant hormone biosynthesis and signalling, which upregulates defence-related genes and enzymes, contributing to reduced postharvest fruit rot and highlighting the advanced strategies citrus plants employ against pathogens [141]. Furthermore, citrus essential oils, rich in linalool and limonene, demonstrate significant antimicrobial effects against *Fusarium* spp. [136]. These essential oils contain phenolic compounds, which exhibit bacteriostatic and fungistatic properties, causing structural damage to fungal cells and thus inhibiting their growth [142]. This antimicrobial prowess of phytochemicals exemplifies an effective defence against a wide array of pathogens.

The deep understanding of citrus defence mechanisms, fuelled by a synergy of genetic, environmental, and biological factors, showcases the antimicrobial strength of phytochemicals. Terpenoids, flavonoids, and alkaloids impede fungal pathogens by disrupting their enzyme systems and cell membrane integrity [143,144]. Recent studies have shown that essential oils from cloves effectively control blue mould in citrus by enhancing defence enzyme activities and stimulating phenylpropanoid metabolism, offering further evidence of the role of phytochemicals in disease control [128,141,142]. These insights encourage ongoing research into leveraging the innate defence capabilities of citrus plants to devise sustainable crop protection strategies. By reducing reliance on synthetic pesticides, this approach aims to promote a more sustainable agricultural ecosystem, demonstrating the potential of natural plant defences in modern agriculture [145,146,147,148].

### 4.6. Exploration of Novel Resistance Strategies

In the citrus industry, efforts are underway to devise innovative strategies for combating *Fusarium* spp., a significant threat to citrus health. One approach involves resistance breeding using rootstocks like Citrus sunki and Rough lemon, known for their inherent resistance to *F. oxysporum* [149]. Further research delves into the mechanisms of nonhost resistance (NHR), particularly in *Citrus limon*’s interactions with *Xanthomonas campestris* pv. *campestris* (Xcc), as investigated by Chiesa et al. [150]. This research highlights how NHR in *C. limon*, triggered by Xcc, results in sustained stomatal closure and involves the salicylic acid (SA) and abscisic acid (ABA) signalling pathways, underscoring the complex interplay of these pathways in plant defence mechanisms. Defence responses in *C. limon* against Xcc include cell wall reinforcement, phenylpropanoid pathway activation, and accumulation of callose and phenolic compounds, pointing to a robust set of mechanisms facilitating NHR. The regulation of the SA signalling pathway and the genes associated with it, alongside the engagement of ABA signalling pathway components, are critical for mounting an effective defence against pathogens like Xcc.

In parallel, studies to screen citrus genetic resources have identified genotypes with resistance to *N. solani*, contributing significantly to our understanding of resistance mechanisms in citrus [151]. This screening is particularly crucial as DRR caused by *F. solani* emerges as a destructive disease in citrus orchards. A pivotal study screening over 1400 citrus germplasm accessions has identified select genotypes such as ‘Fremont’ mandarin and ‘Lamas’ lemon that demonstrate enhanced resistance to *N. solani* [27]. These findings not only inform future research directions but also guide practical disease management applications, suggesting that these identified genotypes may offer promising avenues for further exploration of *N. solani* resistance. This underlines the importance of integrated pest management approaches, combining genetic resistance with an understanding of plant defence mechanisms to safeguard citrus orchards against pathogens and ensure the sustainability of citrus production globally.

## 5. Management Strategies for *Fusarium* spp.

Various methods and approaches are employed to manage soil-borne phytopathogens, including *Fusarium spp*. However, achieving desirable outcomes has proven challenging, primarily due to the intricate interactions between phytopathogens, rhizosphere microbiome and environmental conditions [152]. Currently, no identified cure exists for DDR; therefore, an effective strategy for addressing DRR involves an integrated management approach that combines the cultivation and use of resistant rootstock varieties with adequate sanitation practices, improved cultivation methods, and the utilization of biological control agents and chemical substances [153,154].

### 5.1. Cultural Practises

Prophylactic and sanitation practices serve as preventative measures aimed at eliminating stressors that render trees susceptible to pathogenic attacks. This method plays a crucial role in reducing the introduction and spread of pathogens. Continuous disease monitoring throughout the crop cycle is essential, and the removal of infected roots should be considered. Prudent irrigation methods tailored to the tree’s water needs and ensuring adequate drainage, especially in dense soils, are essential [80]. Rigorous disinfection of equipment, particularly when transitioning between orchards, and the judicious application of fertilizers are imperative for disease control. The recommended course of action for the complete removal of an infected tree is advisable only during the advanced stages of the disease, as suggested by [80]. Solarization is a recommended method, as the elevated temperatures generated can deactivate various soilborne pathogens, rendering *Fusarium* spp. inoculum inactive within a soil temperature range of 42 °C to 47 °C [155].

### 5.2. Chemical Control

Various chemicals have been utilized to manage DRR in citrus; however, there is currently no effective fungicide treatment available [153]. In efforts to mitigate disease incidence, fungicides have been studied against the *Fusarium* spp. complex in several investigations [156]. Fosetylaluminium, propineb, carbendazim, copper oxychloride, thiophanate-methyl, myclobutanil, penconazole, difenoconazole, and mancozeb have demonstrated inhibitory efficacy against *N. solani* mycelial growth in vitro [156]. Farmers combat soilborne pathogens by employing pre-planting soil disinfection techniques, often involving fumigants. Compounds such as Carbendazim, Copper oxychloride, Difenoconazole, Fosetyl-aluminium, Mandipropamid, Metiram, Myclobutanil, Penconazole, Propineb, and Thiophanate-methyl, when applied either in liquid or solid form, inhibit fungal growth, disrupt enzyme activity, or stimulate plant defences specifically targeting pathogens [157].

### 5.3. Phytochemical Control

Currently, there are no successful control measures capable of managing soilborne pathogens. Thus, the necessity of implementing an integrated management strategy arises to address this deficiency [80,158]. Control using phytochemicals is also one of the alternative strategies to chemical control. Among various plant extracts studied under laboratory conditions, including citrus (*C. hystrix*), neem (*Azadirachta indica*), garlic (*Allium sativum*), onion (*Allium cepa*), datura (*Datura stramonium*), calotropis (*Calotropis procera*), peppermint (*Mentha piperita*), fennel (*Foeniculum vulgare*), ginger (*Zingiber officinale*), and chilli (*Capsicum annuum*), higher doses of *A. indica* and *C. procera* extracts demonstrated the most significant inhibition of *N. solani* in vitro mycelial growth, while *C. hystrix* and *C. annuum* extracts were also effective in inhibiting it [157]. Furthermore, the application of commercially available biocides, specifically Plant Guard and Rhizo-N, has proven effective in managing citrus DRR. This effectiveness is manifested in the reduction of disease severity, decreased density of *N. solani* inoculum in the soil, and inhibited colonization of the root system [80]. Furthermore, the efficacy of two silicon-based products in controlling DRR disease in citrus was examined and shown to be a valuable tool in the management of the disease [159]. The application of soil water-soluble silicon emerged as a promising solution for disease management, preventing the deterioration of infected trees and maintaining fruit size, quality, and yield comparable to that of healthy plants.

### 5.4. Biological Control

The use of biological methodologies to manage plant diseases attributed to *Fusarium* spp. is recognized as a promising and environmentally sustainable approach. Numerous research studies have consistently highlighted the effectiveness of various biological control agents (BCAs) in mitigating soil-borne pathogens in controlled environments such as greenhouses and authentic field conditions. BCAs, recognized as promising alternatives for mitigating *Fusarium*-induced damage, have consistently demonstrated commendable efficacy. Species of the *Bacillus* genus exhibit antagonistic properties by producing antibiotics, lipopeptides, or volatile organic compounds (VOCs) that enhance plant resistance, with *Bacillus subtilis* reported as a biocontrol agent against the soil-borne fungus *N. solani* [160,161]. Furthermore, studying the antifungal activity of certain novel peptides isolated from *Bacillus amyloliquefaciens* resulted in the discovery of broad-spectrum antifungal activity against both *N. solani* and *F. oxysporum* [162]. Implementing strategic biological control measures with *B. subtilis* and *Trichoderma* spp. has shown favourable outcomes, leading to a decrease in infection and severity, consequently lowering the population density of *N. solani* in citrus soil [163]. The antagonistic potential of *Trichoderma harzianum* plays a significant role in controlling root and soil diseases caused by nematodes and soil-borne fungi such as *F. oxysporum* [164]. The assessment of biocontrol capabilities of various isolates of *Pseudomonas fluorescens* and *B. subtilis* against pathogens affecting rough lemon (*Citrus × jambhiri* Lush.) showed maximum disease control [165]. Under laboratory conditions, four *P. fluorescens* isolates and two *B. subtilis* isolates demonstrated efficacy in controlling *P. nicotianae* and other citrus phytopathogens, including *Pythium* sp., *Fusarium* sp., and *Colletotrichum gloeosporioides* [165]. Similarly, the inhibitory effects of *B. subtilis* on the mycelial growth of infectious *Fusarium* spp. were shown to be effective in controlling DRR in citrus in greenhouse experiments [153].

The capacity of *Trichoderma* spp. to control dry root rot caused by *F. oxysporum* was investigated, and the study showed a significant decrease in disease incidence in vitro [166]. The in vitro antagonistic activity of *Trichoderma* against *F. oxysporum* was tested on Potato Dextrose Agar medium, while an in vivo greenhouse trial was conducted on Japansche Citroen and Rough Lemon rootstock seedlings. Results indicated that three *Trichoderma* isolates inhibited *F. oxysporum* growth in vitro by over 60%. In glasshouse trials, certain *Trichoderma* isolates reduced the incidence of disease to as low as 3.33% [167]. *Trichoderma* spp. produce secondary metabolites and enzymes such as glucanases, chitinases, and proteases, which disrupt and degrade the cell wall of phytopathogens [168]. Many *Trichoderma* spp. exert their antagonistic activity by competing for nutrients and resources, thereby contrasting colonization by the pathogen and thus making the use of biopesticides a potential alternative to agrochemicals [168]. Although *Trichoderma* spp. are well-known as phytopathogen biocontrol agents, very few commercial bioactive products are currently labelled and widely marketed for use against citrus, with many studies focusing on BCA against *P. digitatum* and no other citrus pathogens, particularly *Fusarium* spp., highlighting a lack of research and representing a problem for citriculture.

### 5.5. Mycoviruses Infecting Fusarium *spp.*

*Fusarium* mycoviruses are ssRNA or double-stranded RNA (dsRNA) viruses with a high specificity for *Fusarium* spp. These mycoviruses thrive within *Fusarium* spp. cells, altering the phytopathogen’s biology. Research into the mycovirus mechanisms reveals a multifaceted scope concerning their ability to influence *Fusarium* spp. pathogenicity, virulence, and overall lifecycle. These mechanisms include antibiosis, where mycoviruses inhibit the growth of pathogenic fungi directly; nutritional competition, where they outcompete the pathogens for resources; and the activation of plant defences, which enhances the plant’s intrinsic resistance to fungal attacks [169,170]. Mycoviruses induce notable changes in *Fusarium* spp. phenotype, affecting growth, spore production, and secondary metabolite biosynthesis. Delving into the mechanisms behind these virus-induced changes suggests both molecular and phenotypic levels of interaction, which could serve as strategic targets to reduce pathogenicity and undesirable morphological transformations in the fungal host [171,172]. Mycovirus transmission methods such as protoplast fusion have been noted to induce hypovirulence in *Fusarium* spp., enhancing the biocontrol possibilities of mycoviruses [171,173,174]. Furthermore, advancements in RNA deep sequencing have expanded our understanding of mycovirus diversity within *Fusarium*, coupled with omics technologies that illuminate the complex interactions between mycoviruses and their fungal hosts. Viruses like FgV1 from the novel Fusariviridae family reduce *F. graminearum* virulence by impacting mycelial growth and mycotoxin production, while FgV-ch9 and FodV1 from the Chrysoviridae family are known to influence fungal host growth, morphology, and pathogenicity on crops such as wheat and maize [175,176]. Employing RNA-Seq for comprehensive transcriptome analysis has illuminated how mycoviruses alter transcriptional regulation and affect phytopathogen host proteins [177,178]. The *FgHEX1* gene in *F. graminearum* is pivotal in asexual reproduction and virulence, also playing a role in viral RNA accumulation and host gene downregulation upon mycovirus infection [179].

Among mycoviruses, those infecting *Fusarium* spp. have garnered attention due to their potential use in biocontrol applications. These mycoviruses, often characterized by ssRNA or double-stranded RNA (dsRNA) genomes, are intricately classified into families by the International Committee for the Taxonomy of Viruses (ICTV), with the most important being Chrysoviridae, Totiviridae, Partitiviridae, and Reoviridae. They exhibit unique intracellular transmission modes, bypassing an extracellular phase and typically infecting hosts within specific vegetative compatibility groups [180,181]. Mycoviruses that reduce the virulence of pathogenic *Fusarium* spp. are particularly promising for developing biocontrol strategies given their potential to affect endophytic fungi and produce killer toxins [182,183,184]. Such mycoviruses, through their hypovirulence, offer a viable strategy for managing *Fusarium* infections, leveraging the unique propagation mechanisms that rely on the fungal host’s lifecycle to spread [171]. Important *Fusarium* spp. in agriculture have been found to host dsRNA mycoviruses like FusoV, which contains dsRNA segments capable of altering the pathogenicity of fungi [174]. This alteration often results in morphological changes and reduced virulence as these mycoviruses disseminate throughout the environment, presenting a compelling case for using mycoviruses in disease management strategies [185]. The discovery of various mycoviruses within *Fusarium* spp., including FgV1, FgV2, and the novel FgNSRV-1, highlights the potential of these viruses to induce phenotypic alterations in their hosts, potentially decreasing pathogenicity and offering a novel approach to controlling *Fusarium*-related diseases [175,186]. The identification of mycoviruses in *F. virguliforme* with reduced virulence and new dsRNA mycoviruses across different *Fusarium* spp. underscores the broader applicability of mycoviruses in citrus disease control strategies [187,188]. Furthermore, research into the interactions between *F. graminearum* and its associated mycoviruses has revealed significant alterations in the host’s gene expression patterns. Notably, genes like *FgHex1* (*FGSG_08737*), a precursor protein crucial for asexual reproduction and virulence, and *FGSG_10089*, a glycosylphosphatidylinositol (GPI)-anchored protein involved in redox-sensing for virulence and cell wall integrity, were found to be upregulated in the presence of mycovirus infection [189]. Conversely, genes encoding putative metabolic enzymes such as enolase (*FGSG_01346*), saccharopine dehydrogenase (*FGSG_10925*), flavohemoglobin (*FGSG_04458*), mannitol dehydrogenase (*FGSG_04826*), and malate dehydrogenase (*FGSG_02504*) were downregulated [185]. Additionally, proteins such as peroxiredoxins (*FGSG_03180* and *FGSG_07536*), which are thought to provide antioxidant protection within the cell, were also found to be downregulated in infected *F. graminearum* strains [190]. This complex interplay of gene expression changes underscores the impact of mycovirus infections on the metabolic and defensive mechanisms within *F. graminearum*.

## 6. Conclusions and Future Prospects

The utilization of molecular-based techniques has revolutionized the identification of *Fusarium* spp. associated with citrus, offering heightened sensitivity and reliability compared to conventional methods. Research indicates a complex involvement of various *Fusarium* and *Neocosmospora* spp. associated with DRR, with recent studies unveiling new species impacting citrus plantations globally. Despite inherent challenges in citrus breeding, the integration of molecular markers and gene pyramiding strategies holds substantial promise for bolstering disease resistance against pathogens such as *Fusarium* spp. Transcription factors (TFs) assume a pivotal role in regulating defence-related gene expression, presenting avenues for genetic engineering and marker-assisted selection (MAS) in cultivating disease-resistant citrus varieties. The orchestration of complex signal transduction networks and subsequent activation of defence-related genes underpin an efficacious immune response. Notably, prophylactic measures encompassing sanitation protocols and refined irrigation practices serve as cornerstones in curtailing pathogen introduction and dissemination. Furthermore, soil disinfection utilizes fumigants alongside the deployment of biological control agents like *B. subtilis* and *Trichoderma* spp. exhibit promising efficacy in reducing *Fusarium* spp. infections.

The *Fusarium*–citrus interaction from molecular and genetic perspectives remains relatively understudied. Advancements in this field of research hold significant potential to deepen our understanding and profoundly influence the cultivation of new germplasms exhibiting heightened tolerance to pathogenic *Fusarium* spp. Enhancements in research methodologies could pave the way for innovative approaches, such as the priming of seeds with beneficial microbes, phytohormones, and natural products, synthetic chemical inducers (chemical priming), as well as nano-emulsions and nanoparticles (nanomaterials priming), to effectively bolster crop resistance against *Fusarium*-linked diseases. Exploiting seed priming, particularly considering advancing knowledge on plant immune memory and priming methodologies, promises novel avenues for fortifying crop protection in future agricultural practices. Moreover, the integration of technologies such as genome-wide screening for defence-related quantitative trait loci (QTLs) in conjunction with CRISPR technology presents an effective strategy for advancing our understanding and fostering the development of resilient plant varieties.

In essence, the early establishment of advanced citrus defence research, coupled with the untapped genetic diversity within citrus accessions, innovative strides in analytics and genomics, and the dynamic interplay of environmental conditions, pathogens, and host genetic backgrounds, are poised to significantly enhance the efficacy and precision of global plant breeding endeavours in the foreseeable future. This trajectory ensures the sustainable growth and advancement of food production and safety on a global scale.

## Figures and Tables

**Figure 1 jof-11-00263-f001:**
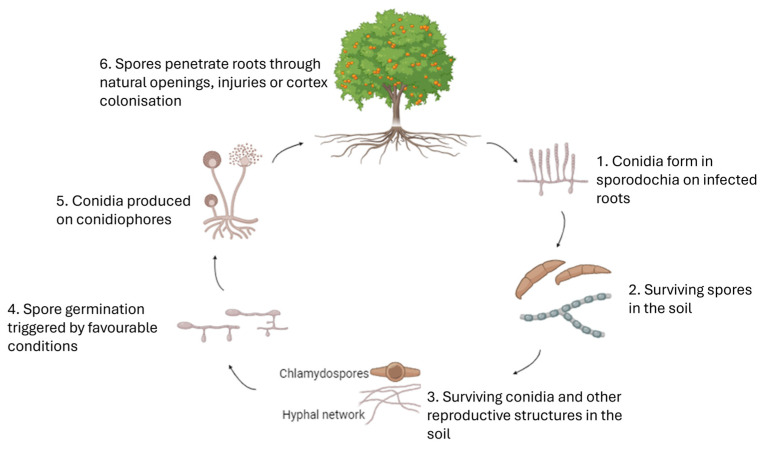
Schematic representation of the progression of *Fusarium* spp. infecting *Citrus* spp. (1) Conidia form in sporodochia on infected roots, where they are produced as asexual spores. (2) Spores persist in the soil, surviving for extended periods, especially in organic matter. (3) Surviving conidia and other reproductive structures, such as chlamydospores and mycelium, act as inoculum sources in the soil. (4) Spore germination occurs when favourable conditions, such as moisture and the presence of roots, are met. (5) New conidia are produced on conidiophores and released into the soil to infect new roots. (6) Spores penetrate roots through natural openings or injuries, leading to cortex colonization and root rot.

**Figure 2 jof-11-00263-f002:**
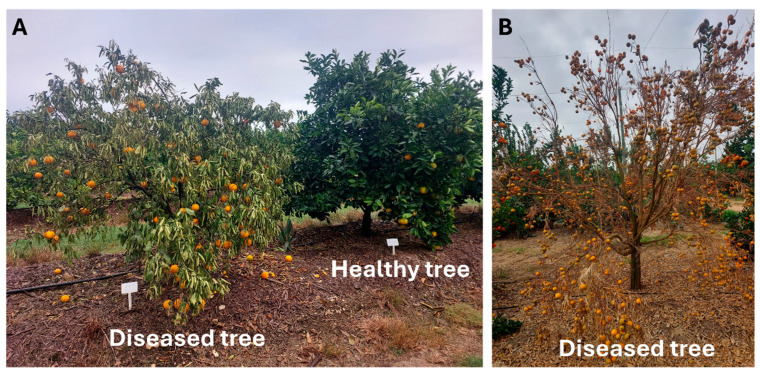
Citrus decline symptoms in a South African orchard: (**A**) Comparison of healthy and diseased citrus trees. (**B**) Severely diseased tree with fruit production.

**Figure 3 jof-11-00263-f003:**
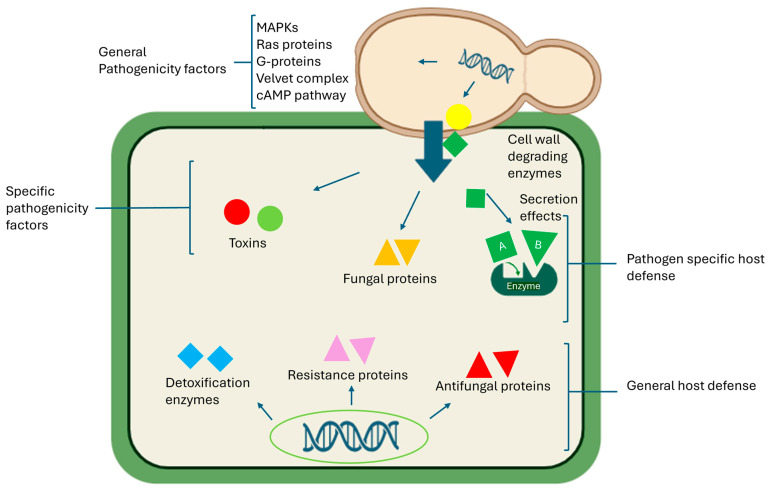
*Fusarium* pathogenicity and host defence mechanisms.

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
