# Peer review of "Fusarium Species Associated with Diseases of Citrus: A Comprehensive Review"

_jof, 2025, doi:10.3390/jof11040263_

Round 1
Reviewer 1 Report
Thank you for giving me this opportunity to review this manuscript. Several Fusarium and Fusarium-like species are important plant pathogens which have been causing serious problems to citrus roots, stems, and also leaves. A systematic review on these fungal species is definitely necessary and useful.
Abstract
The first time appearance of "Fusarium" and "F. solani", it's better to add the name of the mycologist who constructed it.
As you mentioned in the manuscript, "F. solani" is now an abandoned name, please use its updated name and use it consistently through the whole manuscript.
Introduction
I suggest to add several sentences briefly describing the broad Fusarium has been divided into several genera such as Fusarium, Neocosmospora, Bisifusarium, Rectifusarium and why. Or it will be very confusing.
1st paragraph: "several stages of", you can use "nearly all stages of", citrus is very vulnerable to fungal pathogens.
2nd paragraph: When you use "Fusarium" as a taxonomic name, it should be in italic.
"saprobic organisms" to "saprophytes"
3rd paragraph: "these pathogens cause" delete "cause"
"disease causing", may consider use "pathogenic"
2.2
2nd paragraph: A single ITS locus is not enough to classify Fusarium, so it's not necessary to list the results of such studies. You may find some works about how mycologists use multi-loci in identifying Fusarium species, and specify their results. Namely, I suggest to talk about the drawback of using a single ITS locus, and how using multi-loci improved the identification resolution.
2.3 the title, delete "emerging", it means the emergence of new pathogen in phytopathology, obviously you mean the newly identified and outbroken Fusarium pathogens.
Some words like "previously" and "later", I suggest the authors to add the year.
3
This part is not well organized and confusing. The first part should be etiology. Symptomatology is better to be put before epidemiology.
"phytopathology's ability", do you mean "virulence"
4
I have not read many studies on the mechanisms of citrus-Fusarium interaction. So, I think the authors write this part mainly based on other plants-Fusarium interaction, and mainly from cereal plants. It's okay. But the authors should offer the plants' name and make their insights to citrus.
Tables
I know there are many more Fusarium species and caused symptoms on citrus. The authors did not include some other literatures in their summary. I did some reading work on citrus-associated fungi before (2021), and I have my own collection until that time. Here I attach my summary to the authors. I hope it is helpful to the manuscript. By the way, the authors should still check the publications from 2021 to now.
Family | Genus | Species | Distribution | Host | Tissue and symptoms | Culture No. | ITS code | Reference |
Nectriaceae | Bisifusarium | Bisifusarium delphinoides | USA: Florida | Citrus sp. | root: unknown (with soil) | CBS 110316, NRRL 36189 | EU926237 | Schroers et al. 2009 |
Nectriaceae | Bisifusarium | Bisifusarium dimerum | Italy: Reggio | C. medica | fruit: rot | Schroers et al. 2009 | ||
Nectriaceae | Fusarium | F. avenaceum | Iran | Citrus sp. | unknown | FRC R-9369 | NA | Nalim et al. 2009 |
Nectriaceae | Fusarium | F. citri | China: Hunan | C. reticulata | leaf: unknown | CGMCC 3.19467, LC 6896 | MK280803 | Xia et al. 2019; Wang et al. 2019 |
Nectriaceae | Fusarium | F. citricola | Italy: Consenza, Taranto, Vibo Valentia | C. limon, C. reticulata, C. sinensis | crown: canker trunk: canker twig: canker |
CBS 142421, CPC 27805 | LT746245 | Sandoval-Denis et al. 2018 |
Nectriaceae | Fusarium | F. ensiforme | Italy: Catania | C. sinensis | root: rot | CPC 27190 | LT746247 | Sandoval-Denis et al. 2018 |
Nectriaceae | Fusarium | F. equiseti | Pakistan: Faisalabad | C. reticulata | fruit: rot | FUS 21 | MH581300 | Moosa et al. 2021 |
Nectriaceae | Fusarium | F. lateritium | New Zealand | Citrus sp. | CBS 746.79, NRRL 25485 | NA | Sandoval-Denis et al. 2018 | |
Nectriaceae | Fusarium | F. lichenicola | Vietnam | C. maxima | fruit: rot | 2210 | KJ768839 | Amby et al. 2015 |
Nectriaceae | Fusarium | F. oxysporum | Argentina: Tucumán Italy: Catania, Siracusa Tunisia: Bni-Khaled, Manzel Bouzalfa |
C. limon, C. sinensis, C. tangerina | fruit: rot leaf: chlorosis root: rot shoot: wilt |
CPC 27194 | LT746249 | Fogliata et al. 2013; Hannachi et al. 2014; Sandoval-Denis et al. 2018 |
Nectriaceae | Fusarium | F. salinense | Italy: Catania, Messina | C. sinensis | twig: canker | CBS 142420, CPC 26973 | LT746241 | Sandoval-Denis et al. 2018 |
Nectriaceae | Fusarium | F. sarcochroum | Italy: Catania Greece: Missolonghi Spain: Algemesi, Alginet, Castellò |
C. limon, C. reticulata, C. sinensis | trunk: canker twig: dieback |
CPC 26369 | LT746255 | Sandoval-Denis et al. 2018 |
Nectriaceae | Fusarium | F. siculi | Italy: Catania | C. sinensis | root: rot | CBS 142422, CPC 27188 | LT746262 | Sandoval-Denis et al. 2018 |
Nectriaceae | Fusarium | F. sulawense | China: Hunan | C. reticulata | unknown | LC 6897 | MK280810 | Wang et al. 2019 |
Nectriaceae | Fusarium | Fusarium sp. | Tunisia | C. sinensis, C. tangerina | leaf: chlorosis, epinasty shoot: wilt |
P1R1 | KC282838 | Hannachi et al. 2014 |
Nectriaceae | Neocosmospora | Neocosmospora brevis | Italy: Catania | C. sinensis | unknown | CPC 27190 | LT746247 | Sandoval-Denis et al. 2019 |
Nectriaceae | Neocosmospora | Neocosmospora ferruginea | Italy: Siracusa | C. sinensis | root: rot | CPC 28194 | LT746276 | Sandoval-Denis et al. 2019 |
Nectriaceae | Neocosmospora | Neocosmospora macrospora | Italy: Catania | C. sinensis | root: rot | CBS 142424, CPC 28191 | LT746266 | Sandoval-Denis et al. 2018 |
Nectriaceae | Neocosmospora | Neocosmospora martii = Neocosmospora croci |
Italy: Catania | C. sinensis | root: rot | CBS 142423, CPC 27186 | LT746264 | Sandoval-Denis et al. 2018 |
Nectriaceae | Neocosmospora | Neocosmospora phaseoli | Honduras: Puerto Arturo | C. aurantifolia | unknown | BPI 452391 | LR583750 | Sandoval-Denis et al. 2019 |
Nectriaceae | Neocosmospora | Neocosmospora solani = Haematonectria haematococca |
Italy: Catania, Siracusa Philippines: Nueva Vizcaya USA: Texas |
C. aurantium, C. reticulata, C. sinensis | leaf: chlorosis, defoliation root: rot rootstock: dry rot twig: blight |
CPC 27192 | LT746269 | Yago and Chung 2011; Kunta et al. 2015; Sandoval-Denis et al. 2018 |
Nectriaceae | Neocosmospora | Neocosmospora sp. FSSC 9 | Italy: Siracusa | C. sinensis | root: rot | CPC 27195 | LT746275 | Sandoval-Denis et al. 2018 |
Writing
I am not a native English speaker, it is difficult for me to review the details of the language. However, there are two problems of writing that make me feel some paragraphs were confusing.
(1) The description subject was not consistent. For example, the authors aimed to talk about a Fusarium fungus in the first sentence, but in the second sentence, they started description with a caused symptom;
(2) The text was not organized in a specific order, so it's not very fluent to read.
Author Response
Comment 1:
Abstract
The first-time appearance of "Fusarium" and "F. solani", it's better to add the name of the mycologist who constructed it.
As you mentioned in the manuscript, “F. solani" is now an abandoned name, please use its updated name and use it consistently through the whole manuscript.
Introduction
I suggest to add several sentences briefly describing the broad Fusarium has been divided into several genera such as Fusarium, Neocosmospora, Bisifusarium, Rectifusarium and why. Or it will be very confusing.
1st paragraph: "several stages of", you can use "nearly all stages of", citrus is very vulnerable to fungal pathogens.
2nd paragraph: When you use "Fusarium" as a taxonomic name, it should be in italic.
"saprobic organisms" to "saprophytes"
3rd paragraph: "these pathogens cause" delete "cause"
"disease causing", may consider use "pathogenic"
Response 1:
Thank you for your feedback and I acknowledge its value in improving the manuscript. With regards to the abstract, I have implemented your suggestions to the best of my understanding.
With regards to the introductions, I have implemented your suggestions to the best of my understanding by adding a few sentences about the broad Fusarium genus being divided into several genera such as Fusarium, Neocosmospora, Bisifusarium, Rectifusarium.
Paragraph 1,2 and 3 issue has been addressed.
Comment 2:
2.2 2nd paragraph: A single ITS locus is not enough to classify Fusarium, so it's not necessary to list the results of such studies. You may find some works about how mycologists use multi-loci in identifying Fusarium species, and specify their results. Namely, I suggest to talk about the drawback of using a single ITS locus, and how using multi-loci improved the identification resolution.
2.3 the title, delete "emerging", it means the emergence of new pathogen in phytopathology, obviously you mean the newly identified and outbroken Fusarium pathogens.
Some words like "previously" and "later", I suggest the authors to add the year.
Response 2:
(2.2) Referring to the single loci ITS, thank you for your suggestion. I have removed the studies and spoke about the challenges of using single loci markers and the advantages of using multi-loci markers for identification.
(2.3) Thank you for your suggestion. I opted to remove “emerging”. Furthermore, I have add the year linked to the discoveries instead of saying “previously” and “later”.
Comment 3
This part is not well organized and confusing. The first part should be etiology. Symptomatology is better to be put before epidemiology.
"phytopathology's ability", do you mean "virulence"
Response 3:
Thank you for your suggestions and clarifications.
I have changed epidemiology to etiology. Thank you for pointing that out. So now its rather etiology followed by symptomology.
By “phytopathogen’s ability” we are describing how the hyphal network of a phytopathogen spreads through the soil and enhances its ability to infect nearby plants or newly planted trees, In a physical, “reaching” manner. I have therefore amended the sentence to “The hyphal network spreads through the soil, enhancing its ability to infect nearby plants or newly planted trees.” Hopefully that rectifies the misunderstanding.
Comment 4:
I have not read many studies on the mechanisms of citrus-Fusarium interaction. So, I think the authors write this part mainly based on other plants-Fusarium interaction, and mainly from cereal plants. It's okay. But the authors should offer the plants' name and make their insights to citrus.
Responds 4:
Thank you for the suggestion. Yes, citrus-Fusarium interaction is a very poorly published research area. However, I have tried to by more specific and I do mention the plants names and share insights.
Lastly thank you for the table. We will incorporate some of the relevant incidences.

Reviewer 2 Report
This article provides a good summary of the pathogenic Fusarium species associated with diseases of citrus. Although there are many species concluded in Fusarium spp. pathogens related to citrus and and Fusarium is among the most prominent genera in the kingdom fungi, there are few research about the interaction between the these pathogen and citrus have been published. The author wrote a lot basic theory about the interaction of pathogen and host, but very few about the interactions between citrus and Fusarium spp. pathogens have been summerized. so I think this part is not written in depth enough for publishing.
1. The annotated reference number does not correspond to the number cited in the manuscript. For example, third row in Table 2, the root rot caused by Fusarium on Poncirus trifoliata in China is refer to the 39th reference, but not 40th.
2. In the second paragraph of the second page, I think citrus canker is not caused by Fusarium spp. Pathogen. So I suggest that the author should check it again.
Author Response
Comment 1
The annotated reference number does not correspond to the number cited in the manuscript. For example, third row in Table 2, the root rot caused by Fusarium on Poncirus trifoliata in China is refer to the 39th reference, but not 40th.
Response 1:
The suggestion has been implemented.
Comment 2
In the second paragraph of the second page, I think citrus canker is not caused by Fusarium spp. Pathogen. So I suggest that the author should check it again.
Response 2:
The suggestions have been implemented.

Reviewer 3 Report
1. In the introduction, although a large number of references are cited, they are not well integrated with the research objectives of the article. It is suggested that, while citing these studies, a brief summary be included to explain how they align with the theme of the current literature review.
1. "biofuels, medicinal and cosmetics"should be "biofuels, medicine, and cosmetics".
2.“Fusarium”should be"Fusarium".
3. "Fusarium spp. produces"should be"Fusarium spp. produce.Please check and correct all similar cases to ensure the correct agreement between plural subjects and plural verbs.
4. "...Fusarium solani in their citrus orchards [11,24,25,26,27,28,29,30,31,32,33]Furthermore..."should be"...Fusarium solani in their citrus orchards [11,24,25,26,27,28,29,30,31,32,33].Furthermore... ".Please add a period after "[33]".
5. "...Fusarium species are acknowledged... "should be"...Fusarium species are acknowledged... "Please check if all species names in the text are italicized and correct them if necessary.
6.When referring to species, the full name is typically used for clarity. For example, 'Neocosmospora (Fusarium) solani, F.oxysporum, and Fusarium concentricum' should be written as 'Neocosmospora (Fusarium) solani, Fusarium oxysporum, and Fusarium concentricum.'
7. "...usively established. Furthermore, F. sarcochroum, previously documented in Greece [58], was later isolated from wilted lemon and mandarin branches in Italy and [48] ".The placement of "[48]" in [48] seems incorrect and does not properly connect with the previous text. Please check and correct it.
8.Typo in "Fgure 3" . It should be "Figure 3."
9. "...diverse pathogens [112,112,114]. "Please check if there are duplicate citations and correct them.
10.Please correct "trichloronitro methane" to "Trichloronitromethane".
11.The citation number "[189]" in the text does not match the actual number in the reference list. It is recommended to check and make the necessary corrections.
Author Response
Comment 1:
"biofuels, medicinal and cosmetics"should be "biofuels, medicine, and cosmetics".
Response 1:
Comment has been implemented.
Comment 2:
“Fusarium”should be"Fusarium".
Response 2:
Comment has been implemented.
Comment 3:
"Fusarium spp. produces"should be"Fusarium spp. produce.Please check and correct all similar cases to ensure the correct agreement between plural subjects and plural verbs.
Response 3:
Comment and suggestions have been implemented.
Comment 4:
"...Fusarium solani in their citrus orchards [11,24,25,26,27,28,29,30,31,32,33]Furthermore..."should be"...Fusarium solani in their citrus orchards [11,24,25,26,27,28,29,30,31,32,33].Furthermore... ".Please add a period after "[33]".
Response 4:
Comment has been implemented.
Comment 5:
"...Fusarium species are acknowledged... "should be"...Fusarium species are acknowledged... "Please check if all species names in the text are italicized and correct them if necessary.
Response 5:
Comment and suggestions have been implemented.
Comment 6:
When referring to species, the full name is typically used for clarity. For example, 'Neocosmospora (Fusarium) solani, F.oxysporum, and Fusarium concentricum' should be written as 'Neocosmospora (Fusarium) solani, Fusarium oxysporum, and Fusarium concentricum.'
Response 6:
This is particularly a difficult comment to address because from reviewers suggest abbreviated form and some suggest shortened form. There, we have decided to write the species name in full, e.g., Fusarium oxysporum, when first mention in the manuscript or section. Then subsequent mentions in the same context or section: You can abbreviate the genus, e.g., F. oxysporum. We are aware that it can cause confusion with another genus that starts with the same letter (e.g., Fusarium and Fusicoccum), but in that scenario we continue using the full genus name.
Comment 7:
"...usively established. Furthermore, F. sarcochroum, previously documented in Greece [58], was later isolated from wilted lemon and mandarin branches in Italy and [48] ".The placement of "[48]" in [48] seems incorrect and does not properly connect with the previous text. Please check and correct it.
Response 7:
The issue has been corrected.
Comment 8:
Typo in "Fgure 3" . It should be "Figure 3."
Response 8:
The issue has been corrected.
Comment 9;
"...diverse pathogens [112,112,114]. "Please check if there are duplicate citations and correct them.
Response 9:
The issue has been corrected.
Comment 10:
Please correct "trichloronitro methane" to "Trichloronitromethane".
Response 10:
The error has been rectified.
Comment 11:
The citation number "[189]" in the text does not match the actual number in the reference list. It is recommended to check and make the necessary corrections.
Response 11:
All citation number have been checked for correct linking. Thank you

Reviewer 4 Report
This is a comprehensive critical review on diseases of citrus caused by Fusarium and fusarioid (Fusarium-like) species with particular focus on the syndome named Dry Root Rot. This citrus disease is a complax disease, it has a worldwide distribution and a relevant economic importance. In my opinion the review deserves to be published as it provides an updated state of the art.
My major criticisms concern the aspects listed below:
- The general literature on the genus Fusarium, its toxigenic potential and wide host range is poorly cited as is the literature on the application of Trichoderma as a biocontrol agent of plant diseases and the biocontrol of citrus diseases in general.
.- Some formal aspects of the presentation are poorly taken care of. A) troughout the text (including the section References) latin names of genera and species must bee written using italics and in after the first citation of the name of a species the name of the genus can be abbreviated. Conversely the name of fungal genera has to be written using normal characters when it is part of the common name of a disease. The name of the Authors of latin name of a fungus or plant species have to be cited only when the name of a given species is reported for the first time in the text. B) There is a systemic mistake in numbering the references : the number cited in the text does not always correspond to the right article, please check and update the numbers also taking into consideration the references added after the revision.
As the lines of the file are not numbered for detailed comments, please refer to the notes in the text (attached PDF file).
To resume major criticisms see the list below:
- Keywords: add 'Dry Root Rot'
- Subsection 2.2 Rephrase this sentence 'In another study, isolate 34 Fusarium spp. from symptomatic citrus trees in Marocco [21].' It doesen't make sense.
- Subsection 2.3 What do you mean for 'tops'? May be you want to say 'aboveground parts of the plant' or 'tree canopyì.
- Fig 1 Has to be improved (see notes in the text, attached file) as also indicated in the general comments.
- Subsection 4.5 Add reference on phytochemical defence mechanisms in citrus (see note in the text, attached PDF file).
- Section 5 Substitute ' existing microflora in the rhizosphere' with 'rhizosphere microbioma''.
- Subsection 5.2 'chemicals' instead of 'products' and 'pre-planting soil disinfection' instead of 'soil disinfection'.
- Subsection 5.4 Add reference on biological control of citrus disease in general (see note in the text, attached file).
- Subsection 5.5 'concerning' instead of 'revolving around' and 'citrus diseases' instead of 'citricultural diseases'.
- Section 6 Line 4 Add 'and Fusarium-like' (see note in the text, attached PDF file)

Author Response
We sincerely thank you for your thorough evaluation of our manuscript and for providing valuable comments and suggestions. Your insights have been instrumental in improving the clarity and quality of our work.
Comment 1:
Keywords: add 'Dry Root Rot'
Response 1:
The keywords have been amended.
Comment 2:
Subsection 2.2 Rephrase this sentence 'In another study, isolate 34 Fusarium spp. from symptomatic citrus trees in Marocco [21].' It doesn’t make sense.
Response 2:
That section was edited out as another reviewer suggested we rewrite it differently. The section focused on single-loci identification markers and was changed to multi-loci identification markers.
Comment 3:
Subsection 2.3 What do you mean for 'tops'? Maybe you want to say 'aboveground parts of the plant' or 'tree canopy.
Response 3:
Yes, we meant the canopy and we have subsequently changed it to canopy for more clarity.
Comment 4:
Fig 1 Has to be improved (see notes in the text, attached file) as also indicated in the general comments.
Response 4:
Suggestions have been implemented.
Comment 5:
Subsection 4.5 Add reference on phytochemical defence mechanisms in citrus (see note in the text, attached PDF file).
Response 5:
Changes have been implemented.
Comment 6:
Section 5 Substitute ' existing microflora in the rhizosphere' with 'rhizosphere microbioma''.
Response 6:
Suggestion has been implemented.
Comment 7:
Subsection 5.2 'chemicals' instead of 'products' and 'pre-planting soil disinfection' instead of 'soil disinfection'.
Response 7:
Your suggestion was a bit confusing, but we rephrased the sentence to hopefully achieve what you meant.
Comment 8:
Subsection 5.4 Add reference on biological control of citrus disease in general (see note in the text, attached file).
Response 8:
Suggestions have been implemented.
Comment 9:
Subsection 5.5 'concerning' instead of 'revolving around' and 'citrus diseases' instead of 'citricultural diseases'.
Response 9:
Your suggestions have been implemented.
Comment 10:
Section 6 Line 4 Add 'and Fusarium-like' (see note in the text, attached PDF file)
Response 10:
Suggestions have been implemented.

Round 2
Reviewer 1 Report
The authors replied to most of my previous concerns, the manuscript reads much better.
Introduction
p4 & p5, these two paragraphs share some similar sentences, it is better to merge them to avoid repetition.
Author Response
Comments 1: p4 & p5, these two paragraphs share some similar sentences, it is better to merge them to avoid repetition.
Response 1: Thank you for your valuable feedback. I have carefully reviewed the two paragraphs (introduction 4 and 5) and have modified the text to eliminate repetition while maintaining clarity and coherence. The similarities have been adjusted to ensure a smoother flow of information without redundancy. Please let me know if any further refinements are needed.

Reviewer 2 Report
This article provides a good summary of the pathogenic Fusarium species associated with diseases of citrus. Although there are many species concluded in Fusarium spp. pathogens related to citrus and and Fusarium is among the most prominent genera in the kingdom fungi, there are few research about the interaction between the these pathogen and citrus have been published. The author wrote a lot basic theory about the interaction of pathogen and host. It is a good job.
The format of references needs to be unified. For example. pp was used in References 1 "Hazarika, T.K. 2023. Citrus. Singapore: Springer Nature Singapore. In Fruit and Nut Crops 2023, pp. 1-44." However, there are no pp in References 29 "Paguio, O.R. Citrus declinio in the State of Brazil: occurrence and responses to blight diagnostic tests. In International Or ganization of Citrus Virologists Conference Proceedings 1984, 9(1957-2010)".
Author Response
Comments 1: The format of references needs to be unified. For example. pp was used in References 1 "Hazarika, T.K. 2023. Citrus. Singapore: Springer Nature Singapore. In Fruit and Nut Crops 2023, pp. 1-44." However, there are no pp in References 29 "Paguio, O.R. Citrus declinio in the State of Brazil: occurrence and responses to blight diagnostic tests. In International Organization of Citrus Virologists Conference Proceedings 1984, 9(1957-2010)".
Response 1: Thank you for your feedback. We have carefully reviewed all references to ensure uniformity and consistency throughout the manuscript. Please let us know if any further adjustments are required.

Reviewer 3 Report
The author responded well to my questions and made revisions.
N
Author Response
Comments 1: The author responded well to my questions and made revisions.
Response 1: Thank you for your positive feedback. We appreciate your thorough review and have made the necessary revisions based on your suggestions. Please let us know if any further improvements are needed.

Reviewer 4 Report
The Authors addressed almost all criticisms of the reviewer and rephrased sentences as suggested.
Detailed comments refer only to formal aspects suc as the use of normal characters for the genus if it is part of the common name of a disease (for instance Fusarium root rot instead of Fusarium root rot ) or the abbreviated title of journals in the section References. However according with the instructions this is job of the Editors
Author Response
Comments 1: Detailed comments refer only to formal aspects such as the use of normal characters for the genus if it is part of the common name of a disease (for instance Fusarium root rot instead of Fusarium root rot) or the abbreviated title of journals in the section References. However, according with the instructions this is job of the Editors.
Response 1: Thank you for your comment. We have carefully reviewed and corrected all journal abbreviations in the References section and have removed unnecessary italics where applicable. Please let us know if any further adjustments are needed.
